# A protein-truncating R179X variant in *RNF186* confers protection against ulcerative colitis

Manuel A. Rivas[1,2], Daniel Graham[1], Patrick Sulem[3], Christine Stevens[1], A. Nicole Desch[1], Philippe Goyette[4], Daniel Gudbjartsson[3,5], Ingileif Jonsdottir[3,6,7], Unnur Thorsteinsdottir[3,7], Frauke Degenhardt[8], Sören Mucha[8], Mitja I. Kurki[1,2], Dalin Li[9,10], Mauro D'Amato[11,12], Vito Annese[13,14], Severine Vermeire[15,16], Rinse K. Weersma[17], Jonas Halfvarson[18], Paulina Paavola-Sakki[19,20,21], Maarit Lappalainen[19,20,22], Monkol Lek[1,2], Beryl Cummings[1,2], Taru Tukiainen[1,2], Talin Haritunians[9,10], Leena Halme[23], Lotta L.E. Koskinen[22,24], Ashwin N. Ananthakrishnan[25,26], Yang Luo[27], Graham A. Heap[28], Marijn C. Visschedijk[17], UK IBD Genetics Consortium[†], NIDDK IBD Genetics Consortium[‡], Daniel G. MacArthur[1,2], Benjamin M. Neale[1,2], Tariq Ahmad[29], Carl A. Anderson[27], Steven R. Brant[30,31], Richard H. Duerr[32,33], Mark S. Silverberg[34], Judy H. Cho[35], Aarno Palotie[1,2,36,37], Päivi Saavalainen[38], Kimmo Kontula[19,20], Martti Färkkilä[19,20,21], Dermot P.B. McGovern[9,10], Andre Franke[8], Kari Stefansson[3,7], John D. Rioux[4,39], Ramnik J. Xavier[1,25] & Mark J. Daly[1,2]

Protein-truncating variants protective against human disease provide *in vivo* validation of therapeutic targets. Here we used targeted sequencing to conduct a search for protein-truncating variants conferring protection against inflammatory bowel disease exploiting knowledge of common variants associated with the same disease. Through replication genotyping and imputation we found that a predicted protein-truncating variant (rs36095412, p.R179X, genotyped in 11,148 ulcerative colitis patients and 295,446 controls, MAF = up to 0.78%) in *RNF186*, a single-exon ring finger E3 ligase with strong colonic expression, protects against ulcerative colitis (overall $P = 6.89 \times 10^{-7}$, odds ratio = 0.30). We further demonstrate that the truncated protein exhibits reduced expression and altered subcellular localization, suggesting the protective mechanism may reside in the loss of an interaction or function via mislocalization and/or loss of an essential transmembrane domain.

[1] Broad Institute of MIT and Harvard, Cambridge, Massachusetts 02142, USA. [2] Analytic and Translational Genetics Unit, Massachusetts General Hospital, Harvard Medical School, Boston, Massachusetts 02114, USA. [3] deCODE Genetics, Amgen Inc., 101 Reykjavik, Iceland. [4] Research Center, Montreal Heart Institute, Montréal, Québec, Canada H1T1C8. [5] School of Engineering and Natural Sciences, University of Iceland, 101 Reykjavik, Iceland. [6] Department of Immunology, Landspitali, the National University Hospital of Iceland, 101 Reykjavik, Iceland. [7] Faculty of Medicine, University of Iceland, 101 Reykjavik, Iceland. [8] Institute of Clinical Molecular Biology, Christian-Albrechts-University of Kiel, 24118 Kiel, Germany. [9] F. Widjaja Foundation Inflammatory Bowel and Immunobiology Research Institute, Cedars-Sinai Medical Center, Los Angeles, California 90048, USA. [10] Inflammatory Bowel Disease Center, Cedars-Sinai Medical Center, Los Angeles, California 90048 USA. [11] Department of Biosciences and Nutrition, Karolinska Institutet, 14183 Stockholm, Sweden. [12] BioCruces Health Research Institute and IKERBASQUE, Basque Foundation for Science, 48903 Bilbao, Spain. [13] Unit of Gastroenterology, Istituto di Ricovero e Cura a Carattere Scientifico-Casa Sollievo della Sofferenza (IRCCS-CSS) Hospital, 71013 San Giovanni Rotondo, Italy. [14] Strutture Organizzative Dipartimentali (SOD) Gastroenterologia 2, Azienda Ospedaliero Universitaria (AOU) Careggi, 50134 Florence, Italy. [15] Department of Clinical and Experimental Medicine, Translational Research in GastroIntestinal Disorders (TARGID), Katholieke Universiteit (KU) Leuven, Leuven 3000, Belgium. [16] Division of Gastroenterology, University Hospital Gasthuisberg, BE-3000 Leuven, Belgium. [17] Department of Gastroenterology and Hepatology, University of Groningen and University Medical Center Groningen, 9713 GZ Groningen, The Netherlands. [18] Department of Gastroenterology, Faculty of Medicine and Health, Örebro University, SE 701 82 Örebro, Sweden. [19] Department of Medicine, University of Helsinki, 00100 Helsinki, Finland. [20] Helsinki University Hospital, 00100 Helsinki, Finland. [21] Clinic of Gastroenterology, Helsinki University Hospital, 00100 Helsinki, Finland. [22] Research Programs Unit, Immunobiology, and Department of Medical and Clinical Genetics, University of Helsinki, 00014 Helsinki, Finland. [23] Department of Transplantation and Liver Surgery, University of Helsinki, 00100 Helsinki, Finland. [24] Department of Medical Genetics, Biomedicum Helsinki, University of Helsinki, 00100 Helsinki, Finland. [25] Gastroenterology Unit, Massachusetts General Hospital, Harvard Medical School, Boston, Massachusetts 02114, USA. [26] Division of Medical Sciences, Harvard Medical School, Boston, Massachusetts 02114, USA. [27] Wellcome Trust Sanger Institute, Wellcome Trust Genome Campus, Hinxton CB10 1SA, UK. [28] IBD Pharmacogenetics, Royal Devon and Exeter NHS Trust, Exeter EX2 5DW, UK. [29] Peninsula College of Medicine and Dentistry, Exeter PL6 8BU, UK. [30] Meyerhoff Inflammatory Bowel Disease Center, Department of Medicine, School of Medicine, Johns Hopkins University, Baltimore, Maryland, 21205, USA. [31] Department of Epidemiology, Bloomberg School of Public Health, Johns Hopkins University, Baltimore, Maryland, 21205, USA. [32] Division of Gastroenterology, Hepatology and Nutrition, Department of Medicine, University of Pittsburgh School of Medicine, Pittsburgh, Pennsylvania 15261, USA. [33] Department of Human Genetics, University of Pittsburgh Graduate School of Public Health, Pittsburgh, Pennsylvania 15261, USA. [34] Department of Medicine, Inflammatory Bowel Disease Centre, Mount Sinai Hospital, Toronto, Ontario, Canada M5G 1X5. [35] Department of Genetics, Yale School of Medicine, New Haven, Connecticut 06510, USA. [36] Institute for Molecular Medicine Finland, University of Helsinki, 00100 Helsinki, Finland. [37] Massachusetts General Hospital, Center for Human Genetic Research, Psychiatric and Neurodevelopmental Genetics Unit, Boston, Massachusetts 02114, USA. [38] Research Programs Unit, Immunobiology, University of Helsinki, 00100 Helsinki, Finland. [39] Faculté de Médecine, Université de Montréal, Montréal, Québec, Canada H3T 1J4. Correspondence and requests for materials should be addressed to M.A.R. (email: rivas@broadinstitute.org) or to M.J.D. (email: mjdaly@atgu.mgh.harvard.edu).

†A full list of UK IBD Genetics consortium members appears at the end of the paper. ‡A full list of NIDDK IBD Genetics consortium members appears at the end of the paper.

A total of 200 loci have been unequivocally implicated in the two common forms of inflammatory bowel diseases (IBDs): Crohn's disease (CD) and ulcerative colitis (UC)[1,2]. For these findings, like most genome-wide association study (GWAS) results, it has proven challenging to infer the functional consequences of common variant associations[3] beyond cases where protein-altering variants have been directly implicated. Protein-truncating variants (PTVs), also commonly referred to as loss-of-function variants[4] as they often result in a non-functional or unstable gene product, are generally the strongest acting genetic variants in medical genetics and, as one functional copy of the gene is removed, may often provide insight into what is achievable pharmacologically via inhibition of the product of the gene[5]. Thus, identifying PTVs that are demonstrated to lead to loss of gene function and confer protection from disease hold particular promise for identifying therapeutic targets[6–9].

Here we conduct targeted sequencing of the exons of 759 protein-coding genes in regions harbouring common variants associated to IBD in 917 healthy controls, 887 individuals with UC (cases) and 1,204 individuals with CD (cases) to identify predicted PTVs that may confer protection to disease. Through replication genotyping and imputation of a PTV in *RNF186* (p.R179X) in 11,148 UC patients and 295,446 controls we find a significant protective association to UC. By combining RNA allele-specific expression, protein expression and immuno-flourescene imaging, we find that the truncated protein exhibits reduced expression and altered subcellular localization suggesting the protective mechanism may reside in the loss of an interaction or function via mislocalization and/or loss of an essential transmembrane domain.

## Results

**Screen sequencing**. We conducted targeted sequencing of the exons of 759 protein-coding genes in regions harbouring common variants associated to IBD[10,11] in 917 healthy controls, 887 individuals with UC (cases) and 1,204 individuals with CD (cases) from the NIDDK IBD Genetics Consortium (North American clinical samples of European descent). We jointly analysed these data with sequencing data from the same genes taken from an exome-sequencing data set of Finnish individuals: 508 with UC; 238 with CD; and 8,124 Finnish reference samples

sequenced within Sequencing Initiative Suomi (SISu) project (www.sisuproject.fi)[12]. Across this targeted gene set, we discovered 77 PTVs found in 2 or more individuals (Supplementary Table 1), and used a Cochran–Mantel–Haenszel (CMH) $\chi^2$-test to scan for protective variants with two strata corresponding to the two cohorts. The test for association was run based on the phenotype (CD, UC or IBD) indicated by the common variant association in the region[13] (that is, truncating variants in a gene associated only to CD such as at *NOD2* would be tested for CD versus control association). We identified three putatively protective PTVs with a $P$ value $<0.05$: (1) a previously published low-frequency variant in *CARD9* (c.IVS11 + 1G > C) located on the donor site of exon 11, which disrupts splicing ($P = 0.04$)[7,14]; (2) a frameshift indel in *ABCA7* ($P = 0.02$); and (3) a premature stop gain variant (p.R179X) in *RNF186* with signal of association ($P = 0.02$) to UC. As the *CARD9* result was a well-established protective association, and *ABCA7* contained four PTVs that in aggregate did not appear protective (combined odds ratio (OR) = 0.51, $P = 0.21$), we focused specifically on follow-up work to confirm or refute the association of the *RNF186* nonsense variant (the only PTV detected in *RNF186* in either sequence data set).

**Replication**. Replication genotype data obtained in 8,300 UC patients and 21,662 controls from the United States, Canada, the United Kingdom, Sweden, Belgium, Germany, Netherlands and Italy provided strong support that the premature stop-gain allele, p.R179X, confers protection against UC ($P = 0.0028$, OR = 0.36 (95% confidence interval (CI) = 0.19–0.71)). Cluster plots from all genotyping assays were manually inspected to ensure consistent high quality across all experimental modalities used to assess this variant (Supplementary Figs 1 and 2).

Further evidence of replication was seen in whole-genome sequence data followed by imputation collected by deCODE Genetics[15,16], in which a set of 1,453 Icelandic patients with UC were compared with a very large population sample ($n = 264,744$) and a consistent strong protection ($P = 5.0 \times 10^{-4}$, OR = 0.30 (95% CI = 0.15–0.59), imputation information of 0.99; overall replication $P = 8.69 \times 10^{-6}$, OR = 0.33 (0.20–0.55)) was observed between the truncating allele and the disease (Table 1 and Methods). Of note, this observation is advantaged by the property that R179X has a roughly fourfold higher frequency in Iceland (minor allele frequency (MAF) = 0.78%) than in other

**Table 1 | Association of p.R179X in *RNF186* with ulcerative colitis.**

| Study | Data type | UC | | Controls | | Control MAF (%) | P | OR |
|---|---|---|---|---|---|---|---|---|
| | | 179 × | R179 | 179 × | R179 | | | |
| GWASseq | Sequence (targeted) | 0 | 1,774 | 6 | 1,828 | 0.33 | | |
| Finland | Sequence (exome) | 0 | 1,016 | 23 | 16,223 | 0.14 | | |
| Screen | — | 0 | 2,790 | 29 | 18,051 | | 0.022 | 0 |
| US + Canada | Exome chip | 4 | 6,354 | 21 | 12,883 | 0.16 | | |
| UK | Sequencing | 2 | 3,854 | 10 | 7,294 | 0.14 | | |
| Sweden | Exome chip | 2 | 1,518 | 45 | 10,813 | 0.41 | | |
| Belgium | Genotyping | 0 | 1,696 | 0 | 1,764 | 0.00 | | |
| Germany | Genotyping | 1 | 2,035 | 7 | 4,399 | 0.16 | | |
| Dutch | Genotyping | 1 | 1,133 | 8 | 4,164 | 0.19 | | |
| Italy | Genotyping | 0 | 0 | 2 | 1,914 | 0.10 | | |
| Iceland | Sequencing + imputation | 7 | 2,899 | 4,130 | 525,358 | 0.78 | | |
| | | *N* cases | *N* controls | MAF = 0.78% | | | | |
| | | 1,453 | 264,744 | | | | | |
| Replication | | | | | | | $8.69 \times 10^{-6}$ | 0.33 (0.20–0.55) |
| Combined (screen + replication) | | | | | | | $6.89 \times 10^{-7}$ | 0.30 (0.19–0.50) |

Screen + replication *P* value is computed using Mantel–Haenszel $\chi^2$-test with continuity correction.

European populations such that the Icelandic group, despite a moderate contribution in absolute number of cases (close to 1/6), have around half of the contribution in term of effective sample size and power.

Access to imputation data in 150,000 individuals from the Icelandic group enabled us to identify rare loss-of-function homozygotes[17]. We found eight individuals homozygous for the 179X allele, the oldest reached the age of 70 (still alive) and one of the eight died (age 62), consistent with Hardy–Weinberg expectation ($n = 9.1$). There was no significant association of the homozygous genotype with a decreased lifespan or fertility (number of children). Given the lower frequency in other populations, no homozygous individual was expected and/or detected in the Exome Aggregation Consortium (ExAC) nor in the remaining set of individuals in this study. Together this indicates that having two copies with a stop gained in *RNF186* is compatible with life, reproduction and ageing—and importantly does not highlight severe medical consequences that would be of obvious concern in developing a therapeutic to mimic the effect of this allele.

The combined significance across all samples tested is $P = 6.89 \times 10^{-7}$ (OR = 0.30 (95% CI = 0.19–0.50))—considering we advanced only one variant to follow-up study, the replication $P$ value of $8.7 \times 10^{-6}$ is unequivocally significant and would have been significant even if 5,000 variants were put through this follow-up, let alone all 77 that were discovered in the sequencing screen. No other PTV in *RNF186* was discovered and tested in our sequencing. The gene is relatively small (227 amino acids) and R179X is by far the most common detected PTV in ExAC, more than 10 times more common than the sum of all other PTVs in the gene.

*RNF186* is located at 1p36 a locus implicated in a UC GWAS that identified at least two non-coding independent association signals separated by recombination hotspots (rs4654903 and rs3806308; $r^2 = 0.001$, MAF = 45.5% and 47%, respectively) that did not implicate any one of the three genes (*RNF186*–*OTUD3*–*PLA2G2E*) in the region[18]. Recently, a low-frequency coding variant in *RNF186* (rs41264113, p.A64T, MAF = 0.8%) was found to confer increased risk to UC (OR = 1.49 (1.17–1.90))[14]. In the discovery and replication component of this study p.R179X was found to lie on the haplotype background of the non-reference allele for rs4654903, and very little correlation was observed with rs3806308 or the low-frequency coding variant p.A64T (Supplementary Fig. 3). Naturally, the protective signal at p.R179X remains unchanged when corrected for the background allele rs4654903 on the haplotype it arose on (Supplementary Table 2). Furthermore, we did not observe any evidence of association to CD ($P = 0.94$; Supplementary Table 3), consistent with the common variant associations in this region, which are strong and specific to UC only.

**Transcript and protein expression.** *RNF186*, a single-exon protein-coding gene, encodes the ring finger E3 ligase, which localizes at the endoplasmic reticulum and regulates endoplasmic reticulum stress-mediated apoptosis in a caspase-dependent manner[19,20]. To understand the functional consequences of p.R179X we integrated transcriptome and protein expression level data. First, we examined the gene expression profile of *RNF186* (encoded by a single transcript isoform: ENST00000375121) across multiple tissues in the Genotype Tissue Expression (GTEx) project and identify that its highest expression is in the transverse colon (median reads per kilobase of transcript per million mapped reads (RPKM) = 17.32, $n = 61$) with only three other tissues having an RPKM level above 1: (i) pancreas; (ii) kidney cortex; and (iii) the terminal ileum (Supplementary Fig. 4A)[21]. RNF186 protein was observed at 'medium' expression levels for

tissues in the gastrointestinal tract (stomach, duodenum, small intestine, appendix and colon) in the Human Protein Atlas[22] (Supplementary Fig. 4B).

**Impact on transcript allele expression.** We integrated allele-specific expression (ASE) data for individual carriers of p.R179X in the GTEx project. Within GTEx we identified two individual carriers containing ASE data for p.R179X in transverse colon and sigmoid colon. The carriers had consistent patterns of no ASE effects (Supplementary Figs 5 and 6) suggesting that nonsense-mediated decay does not degrade the aberrant transcripts containing the truncating alleles and that additional functional follow-up would be necessary to determine the molecular impact of p.R179X on *RNF186* function (ref. 5). The gene contains one exon and is intronless, and there is prior expectation that those genes do not undergo nonsense-mediated decay since this presence is reported to require the presence of at least one intron.

**Impact on protein allele expression.** Given that R179X messenger RNA was detected at levels similar to the reference allele, we sought to quantify protein expression. Accordingly, we transiently transfected 293T cells with RNF186 expression constructs containing an epitope tag for detection by western blot with anti-V5 antibodies. As expected, we found that the reference allele of *RNF186* was efficiently expressed at the protein level, whereas the truncated protective allele R179X was expressed at reduced levels and the missense risk allele A64T was expressed at higher levels relative to the reference allele (Fig. 1). Notably, the reference allele encodes two transmembrane domains and lacks an N-terminal signal peptide, which supports a model in which RNF186 N and C termini are present on the cytoplasmic side of membrane structures. In contrast, R179X lacks the second transmembrane domain, and must therefore position its N and C termini on opposite sides of the membrane. Collectively, reduced expression and altered membrane topology predict that R179X truncation impairs RNF186 function.

**Impact on cellular localization.** To determine if R179X truncation alters subcellular localization, we over-expressed RNF186 and variant in 293T cells for immunofluorescence imaging. While RNF186 localized to compact intracellular membrane structures, R179X appeared more diffuse, with preferential plasma membrane localization. To directly compare RNF186 and R179X localization in the same cell, we cotransfected 293T cells with Flag-tagged RNF186 and V5-tagged R179X. After staining with anti-V5 and anti-Flag antibodies, we observed very little overlap of RNF186 and R179X localization (Fig. 1). Importantly, these findings suggest that mislocalization of R179X impairs RNF186 function or otherwise alters association with interacting proteins and subsequent ubiquitination of putative substrates.

**Discussion**

This study strengthens the direct evidence for the involvement of *RNF186* in UC risk in which a powerful allelic series, including common non-coding alleles and risk increasing p.A64T is available for further experimentation. Further supporting the medical relevance of this truncating variant, the same rare allele of the same variant coding p.R179X has been reported in Iceland to have a genome wide significant association with a modest increase in serum creatinine level (effect = 0.13 s.d., $P = 5.7 \times 10^{-10}$) and a modest increase in risk of chronic kidney disease[16]. In the context of the few other established protective variants in IBD, including the coding *IL23R* variants (p.V362I, MAF = 1.27%, OR = 0.72 (0.63–0.83); p.G149R, MAF = 0.45%, OR = 0.60 (0.45–.0.79)) and the splice disrupting *CARD9* variant (c.IVS11 + 1G > C, MAF = 0.58%, OR = 0.29 (0.22–0.37))[7],

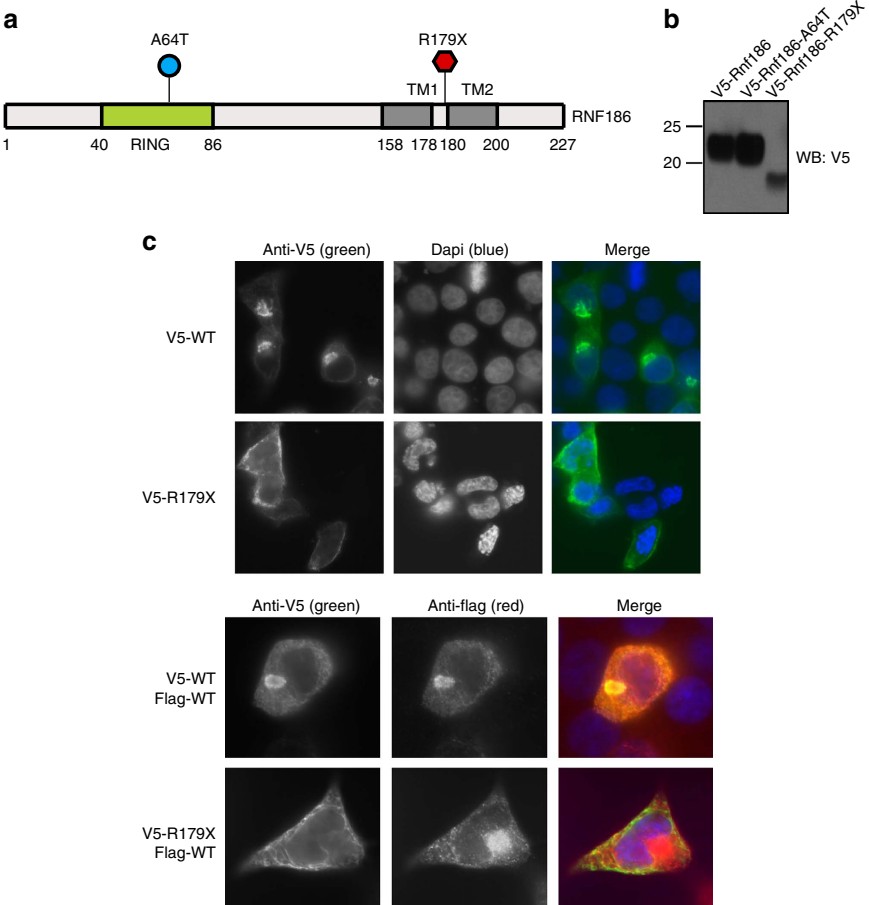

**Figure 1 | R179X impact on protein allele expression and cellular localization.** (**a**) Schematic diagram of the RNF186 protein with a zinc finger RING-type and two helical transmembrane (TM1 and TM2) domains, and the A64T and R179X variants shown. (**b**) 293T cells were transfected with the indicated expression constructs and analysed by western blot for expression of Rnf186 and the R179X variant. The RNF186 protein with a premature stop at amino-acid position 179 is expressed, but at reduced levels. (**c**) 293T cells were transfected with the indicated expression constructs and analysed by immunofluorescence to demonstrate altered subcellular localization of the R179X variant.

R179X exerts a comparable protective effect. Because of the strong protective effect associated with the *RNF186* PTV, studies of RNF186 inhibition and the specific action of this variant protein should be useful in understanding the mechanism by which protection to UC disease occurs and whether this reveals a promising therapeutic opportunity similar to that which has been realized from the example of *PCSK9* and cardiovascular disease.

## Methods
**Ethics statement.** All patients and control subjects provided informed consent. Recruitment protocols and consent forms were approved by Institutional Review Boards at each participating institutions (Protocol Title: The Broad Institute Study of Inflammatory Bowel Disease Genetics; Protocol Number: 2013P002634). All DNA samples and data in this study were denominalized.

**Cohort descriptions.** For all cohorts, UC was diagnosed according to accepted clinical, endoscopic, radiological and histological findings.

Genotyping of the Belgian cohort was performed at the Laboratory for Genetics and Genomic Medicine of Inflammatory (www.medgeni.org) of the Université de Montréal. Belgian patients were all recruited at the IBD unit of the University Hospital Leuven, Belgium; control samples are all unrelated, and without family history of IBD or other immune-related disorders.

NIDDK IBD Genetics Consortium (IBDGC) samples were recruited by the centres included in the NIDDK IBDGC: Cedars Sinai, Johns Hopkins University, University of Chicago and Yale, University of Montreal, University of Pittsburgh and University of Toronto. Additional samples were obtained from the Queensland Institute for Medical Research, Emory University and the University of Utah. Medical history was collected with standardized NIDDK IBDGC phenotype forms. Healthy controls are defined as those with no personal or family history of IBD.

The Prospective Registry in IBD Study at MGH (PRISM) is a referral centre-based, prospective cohort of IBD patients. Enrollment began 1 January 2005. PRISM research protocols were reviewed and approved by the Partners Human Research Committee (#2004-P-001067), and all experiments adhered to the regulations of this review board. The PRISM study data were merged with population controls of European ancestry broadly consented for biomedical studies. These controls included samples from the NIMH repository[23], POPRES[24], the 1000 Genomes Project[25] and controls ascertained for an age-related macular degeneration study[26]. The Italian samples were collected at the S. Giovanni Rotondo 'CSS' (SGRC) Hospital in Italy. The Dutch cohort is composed of UC cases recruited through the Inflammatory Bowel Disease unit of the University Medical Center Groningen (Groningen), the Academic Medical Center (Amsterdam), the Leiden University Medical Center (Leiden) and the Radboud University Medical Center (Nijmegen), and of healthy controls of self-declared European ancestry from volunteers at the University Medical Center (Utrecht).

Subject ascertainment, diagnosis and validation for the UK samples are described elsewhere and are part of the UK Inflammatory Bowel Disease Genetics Consortium (UKIBDGC)[27].

German patients were recruited either at the Department of General Internal Medicine of the Christian-Albrechts-University Kiel, the Charité University Hospital Berlin, through local outpatient services, or nationwide with the support of the German Crohn and Colitis Foundation. German healthy control individuals were obtained from the popgen biobank. Genotyping of the German cohort was performed at the Institute for Clinical Molecular Biology.

Finnish patients were recruited from Helsinki University Hospital and described in more detail previously[28,29].

Subject ascertainment, diagnosis and validation for a subset of the Swedish samples with UC[14] and without UC[30] are described elsewhere.

Icelandic population: a total of 1,453 individuals diagnosed with UC was used in the analysis. All the cases were histologically verified, and diagnosed either by 1997 or prospectively during the period 1997–2009 at Landspitali, the National University Hospital of Iceland.

**NIDDK NHGRI targeted sequencing.** *Sample selection.* We selected 3,008 samples (1,204 CD, 887 UC and 917 controls) for sequencing composed of North-American samples of European descent from the NIDDK IBD Genetics Consortium repository samples.

*Target selection.* Target exonic sequences were selected based on the coding exons of 759 genes (2.546 Mb). Genes were selected if they were in regions identified in the GWASs for inflammatory bowel disease in Franke *et al.* (CD)[10] and Anderson *et al.* (UC)[11].

*Finland exome sequencing.* Finnish individuals were exome sequenced as part of SISu (www.sisuproject.fi). The SISu project consists of the following population and case–control cohorts: 1000 Genomes Project; ADGEN (Genetic, epigenetic and molecular identification of novel Alzheimer's disease-related genes and pathways) Study; The Botnia (Diabetes in Western Finland) Study; EUFAM (European Study of Familial Dyslipidemias); The National FINRISK Study; FUSION (Finland–United States Investigation of NIDDM Genetics) Study; Health 2000 Survey; Inflammatory Bowel Disease Study, METSIM (METabolic Syndrome In Men) Study; Migraine Family Study; Oulu Dyslipidemia Families; Northern Finland Intellectual Disability (NFID); and Northern Finland Birth Cohort (NFBC). All samples were sequenced at the Broad Institute of MIT and Harvard, Cambridge, USA, University of Washington in St. Louis, USA and Wellcome Trust Sanger Institute, Cambridge, UK.

*Exome sequencing.* To produce a harmonized good-quality call set we applied the core variant calling workflow for exome-sequencing data that is composed of two stages that are performed sequentially: pre-processing, from raw sequence reads to analysis-ready reads; and variant discovery, from analysis-ready reads to analysis-ready variants (Supplementary Methods).

*Identification of Finnish samples.* To obtain genetically well-matched controls for comparison with the Finnish IBD cases we first jointly called Finnish exomes with Swedish exomes to identify genetic Finns from other close Nordic country. Final probability was obtained by dividing the probability of being Finnish divided by the sum of probabilities of being Finnish or Swedish. Training samples in distance calculations were selected for being from Finnish or Swedish cohort as appropriate and clustering on the expected cluster (PC1 < 0.002 for Finnish samples and PC1 ≥ 0 and PC2 ≤ 0.01). Samples with ≥ 99% Finnish probability (8,124 non-IBD samples, 508 UC, 238 CD and 92 indeterminate colitis (IC)) were then subset and principal component analysis (PCA) with the same parameters was run again to obtain PCAs for Finnish substructure (Supplementary Figs 7 and 8 and Supplementary Notes).

**Variant annotation.** Variants for the targeted and exome-sequencing data sets were annotated using PLINK/SEQ v0.10 and RefSeq reference transcript set downloaded from https://atgu.mgh.harvard.edu/plinkseq/resources.shtml.

**Follow-up genotyping of RNF186.** *Sequenom.* RNF186 p.R179X was assayed using Sequenom MassARRAY iPLEX GOLD chemistry and SpectroCHIPs were analysed in automated mode by a MassArray MALDI-TOF Compact system 2 with a solid phase laser mass spectrometer (Bruker Daltonics Inc.). The variant was called by real-time SpectroCaller algorithm, analysed by SpectroTyper v.4.0 software and clusters were manually reviewed for validation of genotype calls. Reported genetic map positions for the markers were retrieved from the single-nucleotide polymorphism (SNP) database of the National Center for Biotechnology Information (NCBI).

*Exome array.* The Illumina HumanExome Beadchip array includes 247,870 markers focused on protein-altering variants selected from > 12,000 exome and genome sequences representing multiple ethnicities and complex traits. Nonsynonymous variants had to be observed three or more times in at least two studies, splicing and stop-altering variants two or more times in at least two studies. Additional array content includes variants associated with complex traits in previous GWAS, HLA tags, ancestry informative markers, markers for identity-by-descent estimation and random synonymous SNPs. We focused on variant exm26442, which was the only PTV in the targeted sequencing data set that was also in the exome array and had a *P* value < 0.05 in the screening component of the study. Samples in the targeted sequencing data set were excluded from the exome array analysis.

*UK sequencing.* The UKIBDGC sequenced low-coverage whole genomes of 1,767 UC patients from our nationwide cohort (median depth 2 ×) and compared them with 3,652 population controls from the UK10K project (median depth 7 ×). Samples were jointly called using samtools[31], and subjected to two rounds of genotype improvement using BEAGLE[32]. Genotype count for R179X from exome-sequencing data in 161 additional UK UC patients with severe adverse drug response to common IBD drugs were included.

**Replication in Iceland population.** The Iceland population data have been extended following the step below[15,33].

Sequencing was performed using three different types of Illumina sequencing instruments.

(a) Standard TruSeq DNA library preparation method. Illumina GAIIx and/or HiSeq 2000 sequencers ($n = 5,582$).
(b) TruSeq DNA PCR-free library preparation method. Illumina HiSeq 2500 sequencers ($n = 2,315$).

(c) TruSeq Nano DNA library preparation method. Illumina HiSeq X sequencers ($n = 556$).

Genotyping and imputation methods and the association analysis method in the Icelandic samples were essentially as previously described[15] with some modifications that are described here. In short, we sequenced the whole genomes of 8,453 Icelanders using Illumina technology to a mean depth of at least 10 × (median 32 ×). SNPs and indels were identified and their genotypes determined using joint calling with the Genome Analysis Toolkit HaplotypeCaller (GATK version 3.3.0)[34]. Genotype calls were improved by using information about haplotype sharing, taking advantage of the fact that all the sequenced individuals had also been chip-typed and long range-phased. The sequence variants identified in the 8,453 sequenced Icelanders were imputed into 150,656 Icelanders who had been genotyped with various Illumina SNP chips and their genotypes phased using long-range phasing[35,36].

**Functional consequence of R179X.** *Allele specific expression data for R179X carriers.* The primary and processed data used to generate the ASE analyses presented in this manuscript are available in the following locations: all primary sequence and clinical data files, and any other protected data, are deposited in and available from the database of Genotypes and Phenotypes (www.ncbi.nlm.nih.gov/gap) (phs000424.v6.p1). Tissues with at least eight reads of data are presented.

*Immunofluorescence.* 293T cells were plated on glass coverslips and transfected as described above. Cells were then fixed in 4% paraformaldehyde, blocked (3% BSA, 0.1% saponin, in PBS), and stained with primary antibodies diluted 1:250 in blocking buffer. Primary antibodies were M2 mouse anti-Flag (Sigma F3165-1MG) and rabbit anti-V5 (Cell Signaling Technology D3H8Q). The following secondary antibodies were used at a 1:1,000 dilution in blocking buffer: Alexa Fluor594 goat anti-mouse IgG (Life Technologies R37121) and Alexa Fluor488 goat anti-rabbit IgG (Life Technologies A27034). Cells were mounted in Vectashield medium containing 4,6-diamidino-2-phenylindole (Vector Laboratories) and imaged with a Zeiss Axio A1 microscope equipped with × 63/1.25 objective. Image acquisition was performed with the AxioVision (Rel.4.8) software package.

*Plasmids.* cDNA encoding human RNF186 was obtained from The Genetic Perturbation Platform (GPP, Broad Institute) and cloned by Gibson assembly into the pLX_TRC307 expression construct. Sequences encoding V5 and Flag tags were appended to oligonucleotides for PCR amplification of RNF186.

*Biochemistry.* 293T cells (American Type Culture Collection) were transfected with RNF186 expression constructs by means of Lipofectamine 2000 (Life Technologies) as indicated by the manufacturer. One day after transfection, cells were lysed (1% NP-40 in PBS), resolved by SDS–PAGE, and detected by western blot. Mouse anti-V5 HRP (Sigma V2260-1VL) was diluted 1:5,000 and used in conjunction with chemiluminescent substrate (Pierce SuperSignal West Pico).

**Association analysis.** Association analysis of PTVs in targeted sequencing data and the exome-sequencing data was performed using the CMH $\chi^2$ test implemented in R to screen for PTVs with evidence of protective signal of association[37]. Combined (screen + replication) association analysis was conducted with the CMH $\chi^2$ test. In the replication cohort a set of 1,453 Icelandic patients with UC were compared with a very large group representing the general population ($n = 264,744$). Logistic regression analysis was applied to the data set to obtain study-specific association statistics.

**Data availability.** Raw sequence-based counts of PTVs on which all analyses are based are provided in Supplementary Table 1. Final VCF for the targeted sequencing data set is available on request from NIDDK IBDGC (Phil Schumm <pschumm@uchicago.edu> and Mark J. Daly <mjdaly@atgu.mgh.harvard.edu>).

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

## Acknowledgements

M.J.D. is supported by grants from the following: the National Institute of Diabetes and Digestive and Kidney Disease (NIDDK) and the National Human Genome Research Institute (NHGRI; DK043351, DK064869 and HG005923); the Crohns and Colitis Foundation (3765); the Leona M. & Harry B. Helmsley Charitable Trust (2015PG-IBD001); and Amgen (2013583217). R.J.X. is supported by grants from Amgen (2013583217) and CCFA (3765). J.D.R. is funded by grants from NIDDK (DK064869 and DK062432). IBD Research at Cedars-Sinai is supported by grant PO1DK046763 and the Cedars-Sinai F. Widjaja Foundation Inflammatory Bowel and Immunobiology Research Institute Research Funds. D.P.B.M. is supported by DK062413, AI067068 and U54DE023789-01; grant 305479 from the European Union; and The Leona M. and Harry B. Helmsley Charitable Trust and the Crohn's and Colitis Foundation of America. S.R.B. is support by an NIH U01 grant (DK062431). The sequencing of UK patients was funded by a grant from the Medical Research Council, UK (MR/J00314X/1). The UK10K project was funded by the Wellcome Trust (WT091310). C.A.A. is funded by the Wellcome Trust (098051). J.C. is supported by grants from NIH (U01 DK062429, U01 DK062422, R01 DK092235, SUCCESS). R.H.D. is supported by NIH grant U01 DK062420 and the Inflammatory Bowel Disease Genetic Research Chair at the University of Pittsburgh. We thank the Broad Communications team for the feature image.

## Author contributions

Manuel A Rivas, Mark J Daly, J.D.R, and R.J.X participated in the study design. D.G., A.N.D, and R.J.X provided reagents, designed, and led the functional experiments. C.S. managed the project. P.S., P.G, D.G., I.J, U.T., F.D., S.M., Mitja I. Kurki, D.L, M.A, Vito Annese, S.V, R.K.W, J.H, P.P, M.L, M.L, B.C, T.T, T.H, L.H, L.L.E.K, A.N.A, Yang Luo, G.A.H, M.C.V, D.G.M, B.M.N, Tariq Ahmad, C.A.A, S.R.B, Richard H. Duerr, M.S.S, J.H.C, A.P, P.S, K.K, M.F, D.P.B.M. and Andre Franke provided reagents and tools. Patrick Sulem, D.G, I.J, U.T, and Kari Stefansson generated and analysed the Icelandic replication data. M.A.R led the association analysis. P.G, J.D.R, Mark J. Daly, S.R.B, R.H.D, Mark S. Silverberg, J.H.C, and D.P.B.M are members of the NIDDK IBD Genetics consortium. Carl Anderson is a member of the UK IBD Genetics consortium. All authors commented on the final version of the manuscript. Manuel A. Rivas, D.G., and Mark J Daly wrote the manuscript.

## Additional information

**Competing financial interests:** K.S. and his team report personal fees from deCODE Genetics/Amgen outside the submitted work. The remaining authors declare no competing financial interests.

## UK IBD Genetics Consortium

J. Barrett[28], K. de Lange[28], C. Edwards[40], A. Hart[41], C. Hawkey[42], L. Jostins[43,44], N. Kennedy[45], C. Lamb[46], J. Lee[47], C. Lees[45], J. Mansfield[46], C. Mathew[48,49], C. Mowatt[50], W. Newman[51,52], E. Nimmo[53], M. Parkes[47], M. Pollard[28], N. Prescott[48,49], J. Randall[28], D. Rice[28], J. Satsangi[53], A. Simmons[54,55], M. Tremelling[56], H. Uhlig[57] & D. Wilson[58,59]

[40] Department of Gastroenterology, Torbay Hospital, Devon, UK. [41] Department of Medicine, St. Mark's Hospital, Middlesex, UK. [42] Nottingham Digestive Disease Centre, Queens Medical Centre, Nottingham, UK. [43] Wellcome Trust Centre for Human Genetics, University of Oxford, Headington, UK. [44] Christ Church, University of Oxford, Oxford, UK. [45] Gastrointestinal Unit, Wester General Hospital, University of Edinburgh, Edinburgh, UK. [46] Newcastle University, Newcastle upon Tyne, UK. [47] Inflammatory Bowel Disease

Research Group, Addenbrooke's Hospital, Cambridge, UK. [48] Department of Medical and Molecular Genetics, Guy's Hospital, London, UK. [49] Department of Medical and Molecular Genetics, King's College London School of Medicine, Guy's Hospital, London, UK. [50] Department of Medicine, Ninewells Hospital and Medical School, Dundee, UK. [51] Genetic Medicine, Manchester Academic Health Science Centre, Manchester, UK. [52] The Manchester Centre for Genomic Medicine, University of Manchester, Manchester, UK. [53] Centre for Genomic and Experimental Medicine, University of Edinburgh, Edinburgh, UK. [54] Translational Gastroenterology Unit, John Radcliffe Hospital, University of Oxford, Oxford, UK. [55] Human Immunology Unit, Weatherall Institute of Molecular Medicine, University of Oxford, Oxford, UK. [56] Gastroenterology & General Medicine, Norfolk and Norwich University Hospital, Norwich, UK. [57] Translational Gastroenterology Unit and the Department of Pediatrics, University of Oxford, Oxford, UK. [58] Pediatric Gastroenterology and Nutrition, Royal Hospital for Sick Children, Edinburgh, UK. [59] Child Life and Health, University of Edinburgh, Edinburgh, UK.

## NIDDK IBD Genetics Consortium

C. Abraham[60], J.P. Achkar[61,62], A. Bitton[63], G. Boucher[4], K. Croitoru[64], P. Fleshner[23], J. Glas[63], S. Kugathasan[65], J.V. Limbergen[66], R. Milgrom[35], D. Proctor[60], M. Regueiro[33], P.L. Schumm[67], Y. Sharma[68], J.M. Stempak[35], S.R. Targan[23] & M.H. Wang[32]

[60] Section of Digestive Diseases, Department of Internal Medicine, Yale School of Medicine, New Haven, Connecticut, USA. [61] Department of Gastroenterology and Hepatology, Digestive Disease Institute, Cleveland Clinic, Cleveland, Ohio, USA. [62] Department of Pathobiology, Lerner Research Institute, Cleveland Clinic, Cleveland, Ohio, USA. [63] Division of Gastroenterology, Royal Victoria Hospital, Montréal, Québec, Canada. [64] Inflammatory Bowel Disease Group, Zane Cohen Centre for Digestive Diseases, Mount Sinai Hospital, University of Toronto, Toronto, Ontario, Canada. [65] Department of Pediatrics, Emory University School of Medicine, Atlanta, Georgia, USA. [66] Division of Pediatric Gastroenterology, Hepatology and Nutrition, Hospital for Sick Children, Toronto, Ontario, Canada. [67] Department of Public Health Sciences, University of Chicago, Chicago, Illinois, USA. [68] Genetics and Genomic Sciences, Icahn School of Medicine at Mount Sinai, New York, New York, USA.

