## [Peer Review File · Nature Communications]

Reviewers' comments:

Reviewer #1 (Remarks to the Author):

Authors conducted targeted sequencing of 759 protein coding genes using ulcerative colitis (UC) cases, Crohn's disease (CD) cases, and controls (each for ~1,000 individuals). Authors focused on protein truncating variant (PTV), which should have impact on gene functions. They identified a protective rare variant on UC (R179X in RNF186) was identified. A validation study including ~1,200 UC cases and ~300,000 controls confirmed the protective effect of R179X (OR=0.30). This is one of the initial successes of the large-scale disease genome sequencing study in general populations, and the results looks solid.

One main question is the risk of R179X on CD cases. This reviewer recommends to do novo genotyping of the variant in CD cases, as the authors did in other projects.

Reviewer #2 (Remarks to the Author):

A. Summary of the key results

PTVs are enriched for medically relevant mutations that can potentially provide insight into pharmacological inhibition of the gene, when the mutation leads to loss of function (LOF) that confers protection from disease. The investigators thus deploy an experiment to identify PTVs that are enriched among controls relative to IBD patients. Their filtering algorithm identifies a PTV with strong statistical evidence for protection from UC, being found in only 1.2% of the cases but 1.6% of the controls. Unfortunately, the implicated gene contains only a single exon, so is immune to the mechanism that commonly knocks out function for PTVs (NMD) and biological evidence for this is presented. First, circumvention of NMD is suggested by publically available gene expression data in GTEx. Second, they experimentally demonstrate that the mutation does not inhibit protein expression with the use of a reporter construct. Therefore it cannot be inferred that this PTV is causing a loss of function and no insight is provided to decipher what the mechanism is (e.g. gain of function, dominant-negative).

B. Originality and interest: if not novel, please give references.

The gene RNF186 had previously been implicated in IBD by identification of a missense variant, which mitigates gene-discovery novelty.

C. Data & methodology: validity of approach, quality of data, quality of presentation

Rationale for the experiment is strong, the quality of the genetic data and its presentation are strong. The presentation of the protein expression data is confusing to follow and provides very limited mechanistic insight, as it discounts one of the main assumptions of the method (that PTVs will be LOF).

D. Appropriate use of statistics and treatment of uncertainties

The authors employ appropriate statistics and inferences.

E. Conclusions: robustness, validity, reliability

There is robust statistical evidence for a protective effect of R179X in UC, and biological evidence to exclude NMD. There is no evidence for the biological effect of the variant. Therefore, the suggestion that studies of RNF186 inhibition are warranted by their findings are premature and unsubstantiated

by the presented evidence. Studies to decipher the biological effect of this PTV are clearly needed. The final sentence of the manuscript should be accordingly modified.

F. Suggested improvements: experiments, data for possible revision

I think the paper would be stronger if the other novel protective PTV was not initially excluded and further investigation is included. Specifically after initially identifying 81 PTVs in 2 or more samples, 3 protective PTVs are identified with suggestive evidence for association, two of which had not been previously identified. One of these, in ABCA7, is not further investigated because the gene also "contained four additional PTVs that did not appear protective (combined odds ratio was greater than 1 and a signal of mixture of risk and protective effects would require a sequencing follow-up strategy."

It is not clear that this is a reasonable rationale to disregard the mutation. First, the statistical evidence for the other PTVs in ABCA7 is weak and thus the occurrence of these mutations in two or more samples within their dataset are compatible with chance suggesting that they could bear no etiological relevance. Second, even if some of the risk-conferring PTVs are of etiological significance, these would only enhance our ability to infer 'dose-response' curves from human genetic studies, rendering the gene a more potent provider of pharmacological insight (e.g. PKSC9). Third, it is not clear why a mixture of effects requires sequencing follow-up. If it would be to determine their distributions relative to risk and susceptibility haplotypes, I'm not sure if this is even warranted, given that a lack of correspondence to GWAS-defined haplotypes does not negate biological significance. This is shown in their own data, for example, the protective PTV in RNF186 is correlated with a risk allele from one GWAS SNP and demonstrates no correlation with the other independently association GWAS SNP). Finally, if there is reason to assign all ABCA7 PTVs to haplotypes, they have access to the deCODE Genetics phased sequence data, so if the variants are present in Iceland, it should be relatively straightforward to determine what haplotypes are carrying the PTVs and relatively rapid to do an initial replication of all PTVs in ABCA7.

The paper would be substantially stronger with better investigation of biological evidence for an effect of the mutation; for example protein staining in tissues or a demonstration of apoptotic effects in cell culture.

I think it could help to have a schematic of the RNF186 protein to show the location of the PTV and risk missense variant relative to functional domains.

It is interesting that R179X is also associated with increased serum creatine levels and risk for chronic kidney disease. Is there any clinical link between IBD and kidney function?

G. References: appropriate credit to previous work?

Appropriate credit is given to previous work.

H. Clarity and context: lucidity of abstract/summary, appropriateness of abstract, introduction and conclusions

The report of the protein expression analysis is confusing. It sounds as if the discrepancy in detecting R179X is an experimental artifact, but this is not clear.

Some suggestions for Supplementary Figure 3: It appears that the SNPs are not ordered by physical position, which I find confusing. Also there is no legend to indicate what red and blue represent. For the rare variants, it appears that red is the minor allele. Is the convention the same for the common (allele freq rather than risk). For the GWAS variants, which are risk and which are protective? What data was used in constructing haplotypes? It would help to indicate which GWAS haplotypes are risk

or protective.

Two minor editorial comments, both in the same sentence:

"The gene contains one exon and is intronless, and there is prior expectation that those genes do not undergo nonsense mediated decay [this should be abbreviated as it is explained in the previous sentence] since this presence is reported to require [require?]the presence of at least one intron.

Reviewer #3 (Remarks to the Author):

Rivas et al. report the outcome of a search for protein truncating variants (PTV) associated with IBD risk using a sequencing strategy focused on known common risk loci. In a well designed and well powered genetics experiment, they show clear evidence of a PTV in RNF186 that associates with protection from ulcerative colitis, which they reproduce in a validation cohort.

They also provide some evidence that RNF186 is expressed in both the common and PTV form (using a combination of allele specific expression data and cell line transient transfections).

No other data on the likely functional effects or mechanism of the PTV are reported or addressed.

Rare non-synonymous variants in RNF186 have previously been reported as associated with increased risk of UC by members of the same group.

This study represents a well performed genetics experiment that results in an interesting observation, but is substantially lacking functional/biological data to follow this through. RNF186 has previously been reported to function as an E3 ligase that ubiquitinates targets including BNip1 in response to induction of ER stress. ER stress has previously been implicated in murine models of ileitis (with increasing evidence of a pathogenic role in human Crohn's), but this observation of a potential mutation in an ER stress response gene being associated with protection from UC is potentially very interesting. (See for example human data presented in Treton et al Gastroenterology 2011 and murine data from McGuckin in PLoS 2008). Thus further evaluation of the impact of the PTV on ER stress responses could raise the impact of the manuscript substantially.

As it stands, without further functional data, I think this manuscript lacks interest.

Minor points

There are several minor grammatical errors and awkward phraseology throughout the manuscript which requires improvement.

Page 4 - the 1st paragraph describing the Icelandic cohort needs improvement of the English. The MAF is reported twice at 0.78% - just a few sentences apart (repetition).

REVIEWERS' COMMENTS:

Reviewer #1 (Remarks to the Author):

Authors added the association results of the variant on Crohn's disease risk. This is fine to this reviewer.

Reviewer #2 (Remarks to the Author):

I think that the authors adequately addressed concerns raised by the reviewers. The additional functional data greatly enhances the manuscript and modifications to the text increase the clarity of the report. I recommend publication.

Reviewer #3 (Remarks to the Author):

This review relates to a resubmission of this paper in response to previous comments made by reviewers. I have not repeated my full review here, but address these comments specifically to the additional work performed by the authors in response to concerns that I, and other reviewers, had raised relating to a lack of functional data for the mutation in RNF186 described in the paper.

The authors have added additional data, most notably using transient transfections of wild type and mutant protein into 293T cells using epitope tagging to track sub cellular distribution. In Figure 1, they suggest apparent mislocalisation of the mutant protein.

These data are important in that they provide an attractive hypothesis as to the functional impact of the mutation described in the manuscript. In this regard, they go some considerable way to addressing my concerns. Ideally, I would like to see these explored further, however, with an assessment e.g. of the impact of the mutation on ubiquitination of E3 ligase substrates and/or caspase-3 mediated apoptosis - perhaps using CRISPR/Cas9 to edit endogenous RNF186 in appropriate cell lines.

We would like to thank the reviewers for their positive and constructive feedback – in particular being both gratified that they found the genetic evidence “solid” and “strong” and appreciative of the constructive suggestions for improving the description of molecular function. Below we address the reviewers’ comments in **bold**, and where appropriate we indicate how we have revised the manuscript.

Reviewer 1

Original comment:

Authors conducted targeted sequencing of 759 protein coding genes using ulcerative colitis (UC) cases, Crohn's disease (CD) cases, and controls (each for ~1,000 individuals). Authors focused on protein truncating variant (PTV), which should have impact on gene functions. They identified a protective rare variant on UC (R179X in RNF186) was identified. A validation study including ~1,200 UC cases and ~300,000 controls confirmed the protective effect of R197X (OR=0.30). This is one of the initial successes of the large-scale disease genome sequencing study in general populations, and the results looks solid.

One main question is the risk of R197X on CD cases. This reviewer recommends to do de novo genotyping of the variant in CD cases, as the authors did in other projects.

We would like to thank the reviewer for the remarks on the manuscript. To address the reviewer’s request for additional genetic data quantifying the risk of R179X on CD cases we report CD association result of R179X in Supplementary table 3 (p=0.94; OR=1.04 [95% CI=0.70-1.54]). This is the expected result – that is, consistent with the common variants in this region are strongly associated to UC only – but agree with the reviewer this should be clearly demonstrated.

Reviewer 2

We would like to thank the reviewer for careful read of the manuscript. We have made the requested edits to the manuscript and generated and included additional functional data providing additional support of the loss of function consequence of the truncating 179X allele. We believe that the manuscript is substantially improved thanks to the reviewer’s comments. Below we provide detailed responses to each of the reviewer’s comments and, where appropriate, add references to the changes made in the manuscript.

Original comments:

A. Summary of the key results

PTVs are enriched for medically relevant mutations that can potentially provide insight into pharmacological inhibition of the gene, when the mutation leads to loss of function (LOF) that confers protection from disease. The investigators thus deploy an experiment to identify PTVs that are enriched among controls relative to IBD patients. Their filtering algorithm identifies a PTV with strong statistical evidence for protection from UC, being found in only 1.2% of the cases but 1.6% of the controls. Unfortunately, the implicated gene contains only a single exon,

so is immune to the mechanism that commonly knocks out function for PTVs (NMD) and biological evidence for this is presented. First, circumvention of NMD is suggested by publically available gene expression data in GTEx. Second, they experimentally demonstrate that the mutation does not inhibit protein expression with the use of a reporter construct. Therefore it cannot be inferred that this PTV is causing a loss of function and no insight is provided to decipher what the mechanism is (e.g. gain of function, dominant-negative).

We appreciate the reviewer's comments regarding the requirement to add experimental data to decipher the mechanism. NMD is not a completely efficient cellular mechanism for degrading transcripts containing premature stop codons and not predicted to be at play here, so we completely agree with the reviewer that experimental data is needed to provide insight into which predicted protein truncating variants (commonly referred to as LOF mutations) do indeed lead to loss of function effects, which may be shown through mRNA degradation, protein degradation, or protein staining experiments (localization). To address the reviewer's concern we have added immunofluorescence staining data of 293T cells to assess subcellular localization of Rnf186 and the R179X variant. These experiments demonstrate the mutant RNF186 does not appropriately localize to the compact subcellular structures – in conjunction with the reduced protein expression and lack of a transmembrane domain resulting in N- and C- termini residing on opposite, rather than the same, side of membrane structures – points more clearly to the molecular mechanism of this mutation.

B. Originality and interest: if not novel, please give references.

The gene RNF186 had previously been implicated in IBD by identification of a missense variant, which mitigates gene-discovery novelty.

We thank the reviewer for this comment as it is important to be clear on this point - we had indicated the reference to Beaudoin et al in pg. 4: “Recently, a low-frequency coding variant in *RNF186* (rs41264113, p.A64T, MAF=0.8%) was found to confer increased risk to ulcerative colitis (OR = 1.49 [1.17-1.90])¹⁴.” As the goal of the study was to discover novel protective variants in GWAS hits, the potential existence of a rare risk increasing allele did not immediately suggest the findings presented here.

C. Data & methodology: validity of approach, quality of data, quality of presentation

Rationale for the experiment is strong, the quality of the genetic data and its presentation are strong. The presentation of the protein expression data is confusing to follow and provides very limited mechanistic insight, as it discounts one of the main assumptions of the method (that PTVs will be LOF).

We agree this was not adequately presented and we have rigorously addressed the functional consequences of the 179X allele by incorporating mRNA data, protein expression, and added protein localization data where we can infer the truncating variant has a loss of function consequence.

D. Appropriate use of statistics and treatment of uncertainties

The authors employ appropriate statistics and inferences.

E. Conclusions: robustness, validity, reliability

There is robust statistical evidence for a protective effect of R179X in UC, and biological evidence to exclude NMD. There is no evidence for the biological effect of the variant.

Therefore, the suggestion that studies of RNF186 inhibition are warranted by their findings are premature and unsubstantiated by the presented evidence. Studies to decipher the biological effect of this PTV are clearly needed. The final sentence of the manuscript should be accordingly modified.

The reviewer very accurately summarizes the submission - we have generated and incorporated additional experimental data and more appropriately concluded the manuscript.

F. Suggested improvements: experiments, data for possible revision

I think the paper would be stronger if the other novel protective PTV was not initially excluded and further investigation is included. Specifically after initially identifying 81 PTVs in 2 or more samples, 3 protective PTVs are identified with suggestive evidence for association, two of which had not been previously identified. One of these, in ABCA7, is not further investigated because the gene also "contained four additional PTVs that did not appear protective (combined odds ratio was greater than 1 and a signal of mixture of risk and protective effects would require a sequencing follow-up strategy."

It is not clear that this is a reasonable rationale to disregard the mutation. First, the statistical evidence for the other PTVs in ABCA7 is weak and thus the occurrence of these mutations in two or more samples within their dataset are compatible with chance suggesting that they could bear no etiological relevance. Second, even if some of the risk-conferring PTVs are of etiological significance, these would only enhance our ability to infer 'dose-response' curves from human genetic studies, rendering the gene a more potent provider of pharmacological insight (e.g. PKSC9). Third, it is not clear why a mixture of effects requires sequencing follow-up. If it would be to determine their distributions relative to risk and susceptibility haplotypes, I'm not sure if this is even warranted, given that a lack of correspondence to GWAS-defined haplotypes does not negate biological significance. This is shown in their own data, for example, the protective PTV in RNF186 is correlated with a risk allele from one GWAS SNP and demonstrates no correlation with the other independently association GWAS SNP). Finally, if there is reason to assign all ABCA7 PTVs to haplotypes, they have access to the deCODE Genetics phased sequence data, so if the variants are present in Iceland, it should be relatively straightforward to determine what haplotypes are carrying the PTVs and relatively rapid to do an initial replication of all PTVs in ABCA7.

We thank the reviewer's comment and we have added genetic data quantifying risk of ABCA7 predicted protein-truncating variants. As ABCA7, unlike most genes, harbors many relatively common truncating mutations, we can generate a more powerful composite statistic characterizing truncation variants and in this we see no significant evidence of association.

The paper would be substantially stronger with better investigation of biological evidence for an effect of the mutation; for example protein staining in tissues or a demonstration of apoptotic effects in cell culture.

We thank the reviewer's suggestion and we have included additional experimental data in Figure 1, which we believe bolsters the impact of the study.

I think it could help to have a schematic of the RNF186 protein to show the location of the PTV and risk missense variant relative to functional domains.

This is an excellent suggestion and we have now included the schematic in Figure 1.

It is interesting that R179X is also associated with increased serum creatine levels and risk for chronic kidney disease. Is there any clinical link between IBD and kidney function?

We agree that it is interesting that R179X is also associated with increased serum creatinine levels and risk for chronic kidney disease risk. We are not aware of any evidence for a clinical link between ulcerative colitis and chronic kidney disease and/or serum creatinine levels.

G. References: appropriate credit to previous work?
Appropriate credit is given to previous work.

H. Clarity and context: lucidity of abstract/summary, appropriateness of abstract, introduction and conclusions

The report of the protein expression analysis is confusing. It sounds as if the discrepancy in detecting R179X is an experimental artifact, but this is not clear.

We have improved clarity of the protein expression analysis by describing in detail interpretation of the protein expression data whilst adding protein localization data.

Some suggestions for Supplementary Figure 3: It appears that the SNPs are not ordered by physical position, which I find confusing. Also there is no legend to indicate what red and blue represent. For the rare variants, it appears that red is the minor allele. Is the convention the same for the common (allele freq rather than risk). For the GWAS variants, which are risk and which are protective? What data was used in constructing haplotypes? It would help to indicate which GWAS haplotypes are risk or protective.

We would like to thank the review for these suggestions as it has substantially improved interpretation of the figure. We have made the suggested changes and improved clarity of Supplementary Figure 3 by: 1) adding legend; 2) describing the convention used in the caption; 3) describing the risk and protective GWAS haplotypes; and 4) describing the data set used to construct the haplotypes.

Two minor editorial comments, both in the same sentence:

"The gene contains one exon and is intronless, and there is prior expectation that those genes do not undergo nonsense mediated decay [this should be abbreviated as it is explained in the previous sentence] since this presence is reported to require [require?]the presence of at least one intron.

Thank you. We have made the suggested changes.

Reviewer 3

We would like to thank the reviewer for the insightful comments. We have made the requested edits to the manuscript, and generated and included additional functional data

providing support of the loss of function consequence of the truncating 179X allele. We believe that the manuscript is substantially improved thanks to the reviewer's comments. Below we provide detailed responses to each of the reviewer's comments and, where appropriate, add references to the changes made in the manuscript.

Rivas et al. report the outcome of a search for protein truncating variants (PTV) associated with IBD risk using a sequencing strategy focused on known common risk loci. In a well designed and well powered genetics experiment, they show clear evidence of a PTV in RNF186 that associates with protection from ulcerative colitis, which they reproduce in a validation cohort.

We thank the reviewer for this positive response.

They also provide some evidence that RNF186 is expressed in both the common and PTV form (using a combination of allele specific expression data and cell line transient transfections).

No other data on the likely functional effects or mechanism of the PTV are reported or addressed.

Rare non-synonymous variants in RNF186 have previously been reported as associated with increased risk of UC by members of the same group.

This study represents a well performed genetics experiment that results in an interesting observation, but is substantially lacking functional/biological data to follow this through. RNF186 has previously been reported to function as an E3 ligase that ubiquitinates targets including BNIP1 in response to induction of ER stress. ER stress has previously been implicated in murine models of ileitis (with increasing evidence of a pathogenic role in human Crohn's), but this observation of a potential mutation in an ER stress response gene being associated with protection from UC is potentially very interesting. (See for example human data presented in Treton et al Gastroenterology 2011 and murine data from McGuckin in PLoS 2008). Thus further evaluation of the impact of the PTV on ER stress responses could raise the impact of the manuscript substantially.

As it stands, without further functional data, I think this manuscript lacks interest.

We would like to thank the reviewer for the additional references, which we have now incorporated in the manuscript. Furthermore, in accordance with the reviewer's suggesting we have generated immunofluorescence staining data of 293T cells to assess subcellular localization of Rnf186 and the R179X variant, which supports a loss of function consequence as described above and in the manuscript. We agree with the reviewer that follow-up experiments understanding the role of ER stress in ulcerative colitis is a worthwhile endeavour, but at the moment beyond the scope of this manuscript.

Minor points

There are several minor grammatical errors and awkward phraseology throughout the

manuscript which requires improvement.

Page 4 - the 1st paragraph describing the Icelandic cohort needs improvement of the English. The MAF is reported twice at 0.78% - just a few sentences apart (repetition).

We have made the suggested changes and edited the text accordingly.